# Three-Dimensional Printable Hydrogel Using a Hyaluronic Acid/Sodium Alginate Bio-Ink

**DOI:** 10.3390/polym13050794

**Published:** 2021-03-05

**Authors:** Su Jeong Lee, Ji Min Seok, Jun Hee Lee, Jaejong Lee, Wan Doo Kim, Su A Park

**Affiliations:** 1Medical Device Convergence Center, Konyang University Hospital, Daejeon 35365, Korea; sujeong@kyuh.ac.kr; 2Department of Nature-Inspired System and Application, Korea Institute of Machinery and Materials (KIMM), Daejeon 34103, Korea; jimins@kimm.re.kr (J.M.S.); meek@kimm.re.kr (J.H.L.); 3Department of Nano Manufacturing Technology, Korea Institute of Machinery & Materials (KIMM), Daejeon 34103, Korea; jjlee@kimm.re.kr

**Keywords:** 3D bio-printing, bio-ink, biomaterial, sodium alginate, hyaluronic acid

## Abstract

Bio-ink properties have been extensively studied for use in the three-dimensional (3D) bio-printing process for tissue engineering applications. In this study, we developed a method to synthesize bio-ink using hyaluronic acid (HA) and sodium alginate (SA) without employing the chemical crosslinking agents of HA to 30% (*w*/*v*). Furthermore, we evaluated the properties of the obtained bio-inks to gauge their suitability in bio-printing, primarily focusing on their viscosity, printability, and shrinkage properties. Furthermore, the bio-ink encapsulating the cells (NIH3T3 fibroblast cell line) was characterized using a live/dead assay and WST-1 to assess the biocompatibility. It was inferred from the results that the blended hydrogel was successfully printed for all groups with viscosities of 883 Pa∙s (HA, 0% *w*/*v*), 1211 Pa∙s (HA, 10% *w*/*v*), and 1525 Pa∙s, (HA, 30% *w*/*v*) at a 0.1 s^−1^ shear rate. Their structures exhibited no significant shrinkage after CaCl_2_ crosslinking and maintained their integrity during the culture periods. The relative proliferation rate of the encapsulated cells in the HA/SA blended bio-ink was 70% higher than the SA-only bio-ink after the fourth day. These results suggest that the 3D printable HA/SA hydrogel could be used as the bio-ink for tissue engineering applications.

## 1. Introduction

Three-dimensional (3D) printing techniques have been usually applied in the field of tissue engineering to fabricate a predesigned scaffold, regenerating a defected tissue in brief time [1,2]. In particular, 3D bio-printing systems can fabricate a construct using a hydrogel containing the live cells, i.e., bio-ink, without the cell seeding process for regenerating the defective sites in the body [3]. The hydrogel has been extensively studied in tissue engineering fields because of its potential to provide a suitable environment for the cells in the bio-ink, owing to its high water content, good biocompatibility, and controllable mechanical properties and biodegradability [4]. As one of the requirements of the bio-printing process, regulating the properties of the bio-ink is essential. This can be achieved by varying the types of the base materials composing the bio-ink to obtain the desired biological and physiological properties in various tissues [5]. As a natural polysaccharide, sodium alginate (SA) has been widely used as the base material in bio-printing, owing to its biocompatibility, biodegradability, and viscoelasticity to facilitate cell transplantation by printing [6]. However, SA cannot interact with mammalian cells, owing to its lack of cell adhesion moieties, which is an important requirement for cellular functions [7]. Therefore, studies have proposed that designing the bio-ink by blending various materials can potentially promote cellular activity in bio-inks comprising SA [8,9,10]. Recently, novel tools employing hyaluronic acid (HA) with SA to provide an improved environment to cells were extensively investigated for medical applications [11,12,13]. HA, a glycosaminoglycan that acts as a signaling molecule for cell migration and proliferation, has been extensively used as a hydrogel material in biomedical applications [14,15]. However, it is hard to design a controllable structure with the desired shape, porosity, and interconnected pores in the scaffold for the regeneration of large defects using conventional fabrication methods [16,17]. Therefore, a method for fabricating the scaffold using blended hydrogel and 3D bio-printing needs to be investigated.

In this study, we investigated a biologically enhanced bio-ink using HA, which could improve the cellular functionality of the bio-ink using the alginate alone for 3D bio-printing. We anticipate that SA can support the printability of the bio-ink, while HA enhances the bioactivity in the encapsulated cells. The mechanical properties, such as viscosity, printability, and biocompatibility of the 3D bio-printed scaffolds based on the prepared HA/SA blended bio-ink were analyzed.

## 2. Materials and Methods

### 2.1. Preparation of the HA/SA Bio-Ink

SA was purchased from Sigma-Aldrich (A2033, MO, USA). HA of 2500 kDa was supplied from Humedix.Co. (Anyang, Korea). HA and alginate were dissolved in high glucose-DMEM (Dulbecco’s Modified Eagle’s Medium, Gibco, MD, USA) with 1% penicillin/streptomycin and physically mixed in different ratio (*w*/*v*) in a final concentration of 0%, 10% and 30% (*w*/*v*), respectively. The group names and compositions of the hydrogels are shown in Table 1. The hydrogels were prepared with three different HA/SA volume ratios. For making the bio-ink, the NIH3T3 cells were cultured in high glucose-DMEM supplemented with 10% fetal bovine serum (Gibco, MD, USA) and 1% penicillin/streptomycin (Gibco, MD, USA). These cultures were maintained in the incubator at 37 °C and 5% CO_2_. The cell encapsulated bio-ink gently mixed with HA/SA hydrogels to reach a final concentration of 1 × 10^6^ cells/mL.

### 2.2. Fabrication and Characterization of the HA/SA Scaffold

As shown in Figure 1, we used a 3D solid freeform fabrication (SFF) bio-printing system (laboratory made system in the Korea Institute of Machinery and Materials) [7]. We fabricated a cell-laden construct in a layer-by-layer fashion with the designed 20 × 20 × 1.5 mm^3^. 

HA/SA hydrogels were analyzed by the rheometer (TA Instruments, New Castle, DE, USA) at the room temperature to measure the viscosities of the hydrogels. The infrared spectra of HA/SA hydrogel were measured using an FT-IR spectrophotometer (Perkin-Elmer, Waltham, MA, USA) measured spectrum condition at a resolution of 4.0 cm^−1^ over the range of 500–4000 cm^−1^. For the frequency sweep experiments, frequency was varied from 1 to 10 Hz at 5% strain while for the strain sweep experiments. The printability of the HA/SA hydrogels were evaluated by bio-printing system. To assess shrinkage behavior, the hydrogel printed structures were measured for size variation before and after treatment with CaCl_2_ (Sigma-Aldrich, St. Louis, MO, USA). The shrinking rate analysis of structure area was carried out using an image J program (NIH, Bethesda, MA, USA). 

### 2.3. Cytocompatibility of the Cultured Cells in HA/SA Scaffold

For cell viability analysis, the encapsulated cells in the constructs were evaluated using a Live/Dead Cell assay kit (Molecular Probes, Invitrogen, Carlsbad, CA, USA) according to the manufacturer’s instructions for 7 days. All the constructs were washed with PBS (phosphate buffered saline, Gibco, Gaithersburg, MD, USA), moved to another 24 well plate. The samples were stained for 20 min in the dark and washed 3 times in PBS. Live and dead cells were observed using fluorescence microscopy (Nikon, Tokyo, Japan). Furthermore, the cell-laden constructs were cut on 2.4 × 2.4 mm^2^ size and then analyzed with WST-1 (Life Technologies, Carlsbad, CA, USA) to analyze the cell proliferation for 4 days. Briefly, the constructs were washed with PBS, moved to another 48 well plate. In the next, 40 μL of WST-1 reagent and 400 μL of serum free medium were added in each well and then incubated in the dark for 40 min at 37℃. After incubation, 100 μL of the incubated medium was transferred to a 96 well plate, and the absorbance at 450 nm was immediately measured using a microplate spectrophotometer (Bio-Rad, Hercules, CA, USA). The cell viability was examined after incubation for five samples per each group, and the WST-1 reagent with serum free medium was used as the blank control.

### 2.4. Statistical Analysis

All experiments were carried out at least three times and one-way analysis of variance was conducted for statistical analyses at the significance level of *p* < 0.05. 

## 3. Results and Discussion

### 3.1. Fabrication and Characterization of the HA/SA Bio-Ink 

The SA is extensively used as the basement material of the bio-ink in the tissue engineering field because their physical and mechanical properties satisfy the requirements of an ‘ideal’ bio-ink [18]. As shown in Figure 2A, the viscosity of the blended bio-ink was visually increased with HA before pre-crosslinking. Moreover, these bio-inks after crosslinking were measured using rheometer to evaluate about their quantitative viscosity. S100H0, S90H10, and S70H30 had viscosities of 883, 1211, and 1525 Pa∙s, respectively, at a 0.1 s–1 shear rate (Figure 2B). In particular, all of the bio-inks were shown the shear thinning behavior decreasing shear viscosity with increasing shear rate. As the results, the viscosity of the bio-ink increased with HA, which satisfied with the condition to encapsulate the cells in the hydrogel and could be matched with the elasticity of the various soft tissue [19]. 

To characterize the chemistry of the HA/SA blends, we analyzed the pre-crosslinked hydrogels by using FT-IR. As shown in Figure 3, the –OH stretching vibrations were observed at 3255.88 cm^−1^ in S100H0, which shifted with adding HA at 3246.46 cm^−1^ in S70H30. The alkane –CH stretching vibrations were measured at 2883.43 cm^−1^ in S100H0, which also shifted with adding HA at 2927.59 cm^−1^ in S70H30. The asymmetric and symmetric stretching of the carboxyl group of SA at 1605.26 and 1404.79 cm^−1^ in S100H0 were shifted with adding HA at 1598.44 and 1408.80 cm^−1^ in S70H30, respectively [20,21]. These results were consistent with the analysis of the viscosity and bio-inks were successfully blended according to the ratio of HA/SA. Based on these results, we printed the bio-ink using 3D bio-printer. 

### 3.2. Fabrication and Characterization of the HA/SA Scaffold

The scaffolds were printed after preparation and characterization of the bio-ink. In Figure 4, the all of bio-inks were completely plotted with maintained their structure from the nozzle at the printing head in the same printing conditions. In particular, the pre-crosslinked SA hydrogel with 1% CaCl_2_ was used for appropriate printing, and then, the scaffolds successfully fabricated with stacked layer-by-layer without any crosslinking of the HA. Even though the size of the strands increased compared to SA-only bio-ink, the scaffolds maintained their porous structure after stacking process. These results indicated the pre-crosslinking rate dependent on the ratio of SA has an important role to construct the strut in the scaffold in contrast to the viscosity results. Furthermore, as shown in Figure 5, post-crosslinking process using with 5% CaCl_2_ was carried out for fully cross-linked SA to stable structure, which could prevent collapse entire structure in early time. One of the major challenges for successful tissue regeneration, the scaffold must maintain the structure during application periods [22]. Although shrinkage rate decreased with the ratio of HA in the bio-ink, we anticipate that the structural stability of the scaffold could better with the ratio of SA. 

### 3.3. Cytocompatibility of the Cultured Cells in HA/SA Scaffold

Cell viability and the cell proliferation of the scaffolds using Live/Dead staining and WST-1 assay, respectively. As shown in Figure 6, the scaffold maintains their structure in the S100H0 and S90H10 scaffold for 7 days although the degradation of the scaffold was shown in S70H30 scaffold at 7 day. This result means the structural stability by crosslinking rate of SA have a critical role in the bio-printed scaffold. However, the encapsulated cells were successfully live in the all groups, especially, the viability of the cells increased with the ratio of the HA in the bio-ink. 

Furthermore, in Figure 7, the cell proliferation property of the scaffolds increased for 4 days. There was no significant difference between S100H0 at day 1 and day 4. However, the cell proliferation of the scaffolds increased until 4 days in the case of the S90H10 and S70H30 scaffolds. These results mean HA could enhance the viability of the encapsulated cells with biological interactions of extracellular matrix [23]. In particular, one of important criterion to print the cell-laden bio-ink is that the bio-ink should come out of the nozzle with minimum applied shear force may damage the cells and reduce the cell viability in the printed constructs [24]. In this work, we suggest the blended HA could improve a protective function of bio-ink than SA-only. Therefore, the HA could positively effects on the cell viability and proliferation of the scaffold although it weakens the crosslinking of the scaffolds because of the HA ratio in the bio-ink that was not cross-linked. Based on the results, we anticipate the HA/SA blended hydrogel could be used as a bio-ink for enhancing cell activities by regulating the ratio of the HA in tissue engineering field and biomedical applications. 

## 4. Conclusions

In this study, we developed the HA/SA blended hydrogel bio-ink for 3D bio-printing application. The viscosity of the bio-inks was regulated HA with pre-crosslinking treatment using CaCl_2_, which also well-blended physically according to the ratio of the hydrogels. In particular, the cell viability and proliferation increased with adding the ratio HA in the bio-ink although that were more disintegrated than SA-only scaffold. Furthermore, we could improve the bio-ink properties by applying the motif to enhance the binding affinity with cells on SA hydrogel and increasing the crosslinking property of HA to enhance the cell viability as well as the structural stability. Consequently, we suggest that these blended 3D printable HA/SA hydrogel potentially could be a good candidate for the application of the soft tissue regeneration. 

## Figures and Tables

**Figure 1 polymers-13-00794-f001:**
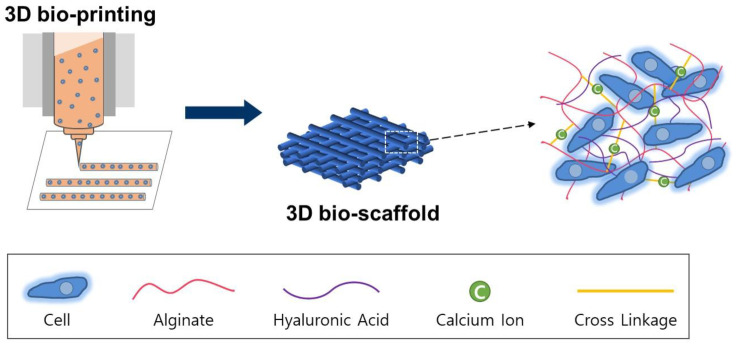
Schematic illustration of hyaluronic acid and alginate mixture as bio-ink for 3D bio-printing.

**Figure 2 polymers-13-00794-f002:**
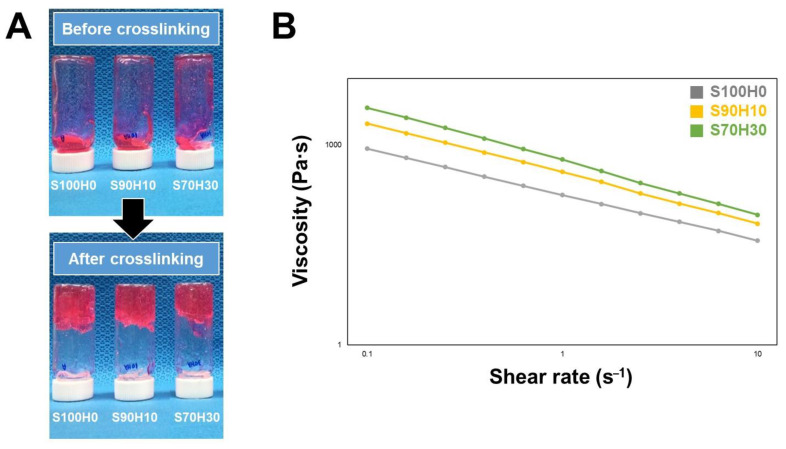
Viscosity of the bio-ink hyaluronic acid and alginate mixture as bio-ink for 3D bio-printing. (**A**) Blended bio-inks before pre-crosslinking (top) and after pre-crosslinking (bottom) with 1% CaCl_2_ (**B**) Viscosity of the pre-crosslinked bio-inks.

**Figure 3 polymers-13-00794-f003:**
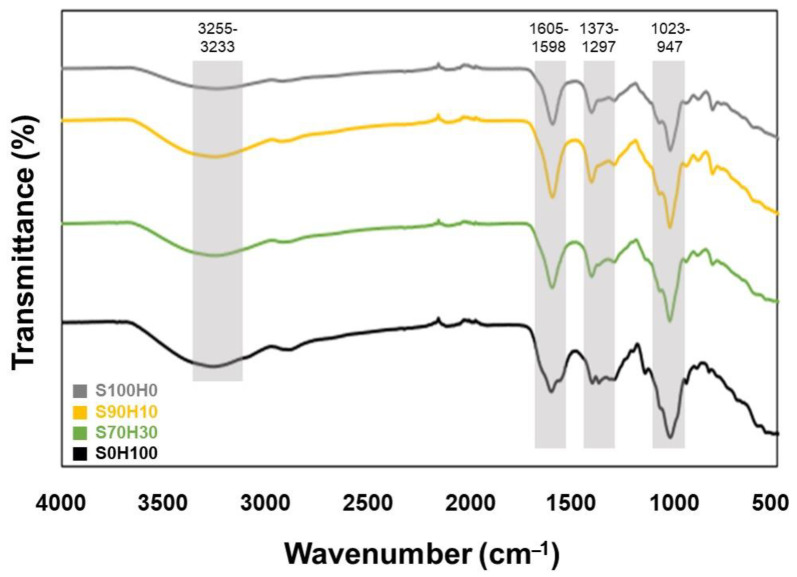
FT-IR spectrum of the bio-inks.

**Figure 4 polymers-13-00794-f004:**
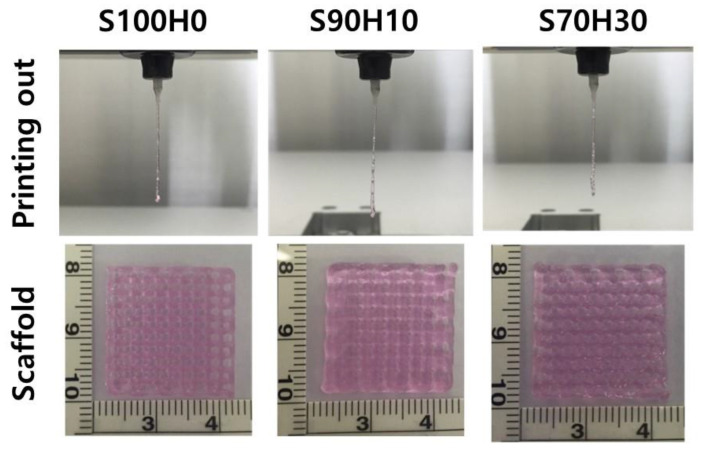
Printability of the bio-inks.

**Figure 5 polymers-13-00794-f005:**
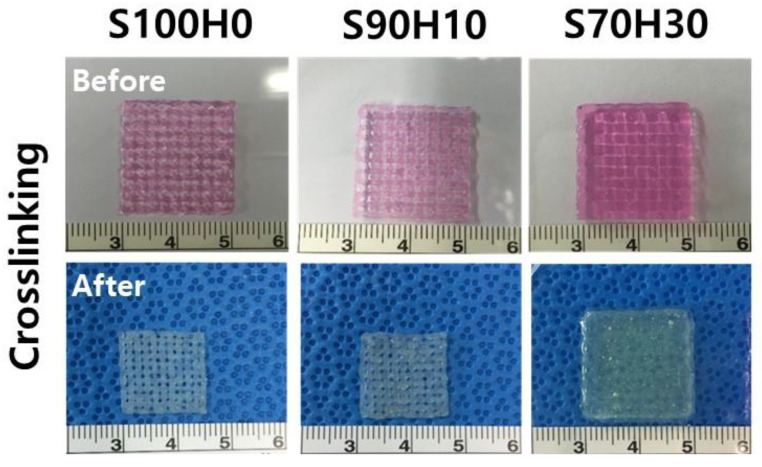
Structure of the scaffolds after crosslinking with 5% CaCl_2_.

**Figure 6 polymers-13-00794-f006:**
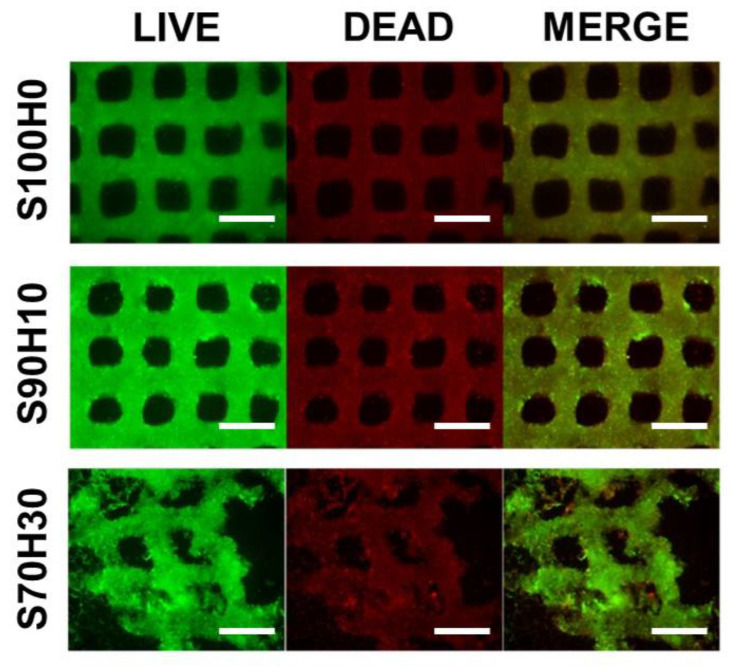
Cell viability of the encapsulated cells in the scaffolds. (scale bar = 1000 μm).

**Figure 7 polymers-13-00794-f007:**
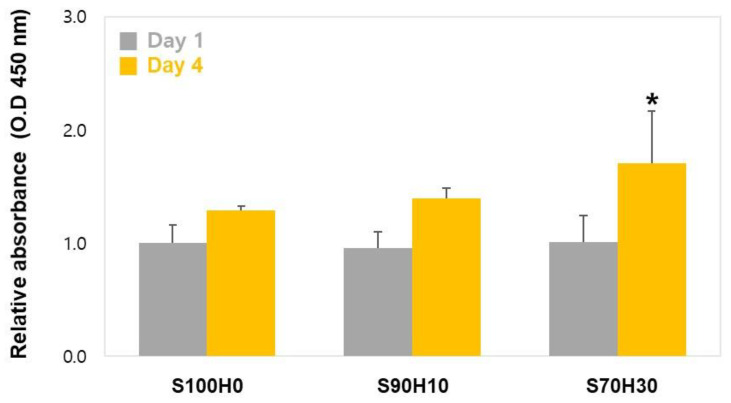
Proliferation of the encapsulated cells in the scaffolds. (Normalized at S100H0, Day 1, n = 3, * *p* < 0.05, asterisks (*) was compared to S100H0).

**Table 1 polymers-13-00794-t001:** Acronyms and compositions of the sodium alginate and hyaluronic acid blends.

Acronym	Composition (%, *v*/*v*)	
	Sodium Alginate (S)	Hyaluronic Acid (H)
S100H0	100	0
S90H10	90	10
S70H30	70	30

## Data Availability

The data presented in this study are available on request from the corresponding author.

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
