# Peer review of "Three-Dimensional Printable Hydrogel Using a Hyaluronic Acid/Sodium Alginate Bio-Ink"

_polymers, 2021, doi:10.3390/polym13050794_

Round 1
Reviewer 1 Report
Dear Author,
Authors’ team in manuscript polymers-1129768, entitled “Three-Dimensional Printable Hydrogel using a Hyaluronic Acid/Sodium Alginate Bio-ink" showed a method to fabricate a blended bio-ink based on hyaluronic acid and sodium alginate, HA/SA, without chemical crosslinker. The obtained HA/SA bio-ink was characterized by viscosity, printability, and shrinkage properties for their suitability in bio-printing, as well for their biocompatibility.
The abstract is descriptive and qualitative, provides briefly the purpose of the manuscript as well as meaningful conclusions.
Introduction is well organized and provide clear, short description of the latest developments in this field, the goals of manuscript, methods analyzed and expected results. The authors could consider adding the following articles in analysis:
- Antich, C., de Vicente, J., Jiménez, G., Chocarro, C., Carrillo, E., Montañez, E., ... & Marchal, J. A. (2020). Bio-inspired hydrogel composed of hyaluronic acid and alginate as a potential bioink for 3D bioprinting of articular cartilage engineering constructs. Acta biomaterialia, 106, 114-123.
- Song, S. J., Choi, J., Park, Y. D., Hong, S., Lee, J. J., Ahn, C. B., ... & Sun, K. (2011). Sodium alginate hydrogel‐based bioprinting using a novel multinozzle bioprinting system. Artificial organs, 35(11), 1132-1136.
- Puertas-Bartolomé, M., Włodarczyk-Biegun, M. K., Del Campo, A., Vázquez-Lasa, B., & San Román, J. (2020). 3D Printing of a Reactive Hydrogel Bio-Ink Using a Static Mixing Tool. Polymers, 12(9), 1986.
I would suggest to break folder 2. Materials and Methods (Lines 51-65) into two separated subfolders as: 2.1. Materials (with provided necessary data for all used chemicals, e.g. Manufacturer, City, State, Country) and 2.2. Fabrication of HA/SA scaffold (with provided necessary data for all used devices, e.g. Type, Manufacturer, City, State, Country).
It is needed to insert FTIR spectra of used hyaluronic acid and sodium alginate before 3D bio-printing in Figure 3, as well as FTIR spectra of bio-ink before and after crosslinking, for better comparison and analysis results of obtained blends.
Could the authors provide data about the amount of unreacted hyaluronic acid and sodium alginate, which is also important for in vivo application?
Did the authors analyze swelling properties in different physiological fluids (e.g. at pH 7.0 and pH 7.4 and 37 °C)? The swelling ratio measurement will be useful for better characterization and possible future application in soft tissue regeneration.
Also, I would like to advise authors for additional improvement of this paper quality by adding morphological analysis of obtained bio-ink, lyophilized in swollen state, e.g. by using scanning electron microscope method.
All figures and table should be inserted in the main text according Instruction and Journal`s template.
It is common for scientific papers to be written in passive. Please, avoid 1st person plural and rewrite all sentences to 3rd person plural (Lines: 14, 42, 44, 62, 110, 151, 155, 159 and 164).
After revision by the authors, this manuscript can be considered to be published preferably as Short Communication in the journal Polymers.
Best regards,
Reviewer
Author Response
- Suggestion 1
The abstract is descriptive and qualitative, provides briefly the purpose of the manuscript as well as meaningful conclusions.
- Response
We appreciate your kind comments. According to the mentioned suggestion, we have included more quantitative results and descriptions in the text as below,
“Bio-ink properties have been extensively studied for use in the three-dimensional (3D) bio-printing process for tissue engineering applications. In this study, we developed a method to synthesize bio-ink using hyaluronic acid (HA) and sodium alginate (SA) without employing the chemical crosslinking agents of HA to 30% (w/v). Furthermore, we evaluated the properties of the obtained bio-inks to gauge their suitability in bio-printing, primarily focusing on their viscosity, printability, and shrinkage properties. Furthermore, the bio-ink encapsulating the cells (NIH3T3 fibroblast cell line) was characterized using a live/dead assay and WST-1 to assess the biocompatibility. It was inferred from the results that the blended hydrogel was successfully printed for all groups with viscosities of 883 Pa∙s (HA, 0% w/v), 1211 Pa∙s (HA, 10% w/v), and 1525 Pa∙s, (HA, 30% w/v) at a 0.1 s−1 shear rate. Their structures exhibited no significant shrinkage after CaCl2 crosslinking and maintained their integrity during the culture periods. The relative proliferation rate of the encapsulated cells in the HA/SA blended bio-ink was 70% higher than the SA-only bio-ink after the fourth day. These results suggest that the 3D printable HA/SA hydrogel could be used as the bio-ink for tissue engineering applications.”
- Suggestion 2
Introduction is well organized and provide clear, short description of the latest developments in this field, the goals of manuscript, methods analyzed and expected results. The authors could consider adding the following articles in analysis:
- Antich, C., de Vicente, J., Jiménez, G., Chocarro, C., Carrillo, E., Montañez, E., ... & Marchal, J. A. (2020). Bio-inspired hydrogel composed of hyaluronic acid and alginate as a potential bioink for 3D bioprinting of articular cartilage engineering constructs. Acta biomaterialia, 106, 114-123.
- Song, S. J., Choi, J., Park, Y. D., Hong, S., Lee, J. J., Ahn, C. B., ... & Sun, K. (2011). Sodium alginate hydrogel‐based bioprinting using a novel multinozzle bioprinting system. Artificial organs, 35(11), 1132-1136.
- Puertas-Bartolomé, M., Włodarczyk-Biegun, M. K., Del Campo, A., Vázquez-Lasa, B., & San Román, J. (2020). 3D Printing of a Reactive Hydrogel Bio-Ink Using a Static Mixing Tool. Polymers, 12(9), 1986.
- Response
We appreciate your kind comments. According to the mentioned suggestion, we have included more descriptions with recommended references in the text as below,
“Therefore, studies have proposed that designing the bio-ink by blending various materials can potentially promote cellular activity in bio-inks comprising SA [8-10].”
“However, it is hard to design a controllable structure with the desired shape, porosity, and interconnected pores in the scaffold for the regeneration of large defects using conventional fabrication methods [16, 17].”
References
[10] Antich, C.; de Vicente, J.; Jiménez, G.; Chocarro, C.; Carrillo, E.; Montañez, E.; Gálvez-Martín, P.; Marchal, J.A. Bio-inspired hydrogel composed of hyaluronic acid and alginate as a potential bioink for 3D bioprinting of articular cartilage engineering constructs. Acta Biomater. 2020, 106, 114–123, doi:10.1016/j.actbio.2020.01.046.
[16] Song, S.J.; Choi, J.; Park, Y.D.; Hong, S.; Lee, J.J.; Ahn, C.B.; Choi, H.; Sun, K. Sodium Alginate Hydrogel-Based Bioprinting Using a Novel Multinozzle Bioprinting System. Artif. Organs 2011, 35, 1132–1136, doi:10.1111/j.1525-1594.2011.01377.x.
[17] Puertas-Bartolomé, M.; Włodarczyk-Biegun, M.K.; Del Campo, A.; Vázquez-Lasa, B.; Román, J.S. 3D Printing of a Reactive Hydrogel Bio-Ink Using a Static Mixing Tool. Polymers (Basel). 2020, 12, 1–17, doi:10.3390/polym12091986.
- Suggestion 3
I would suggest to break folder 2. Materials and Methods (Lines 51-65) into two separated subfolders as: 2.1. Materials (with provided necessary data for all used chemicals, e.g. Manufacturer, City, State, Country) and 2.2. Fabrication of HA/SA scaffold (with provided necessary data for all used devices, e.g. Type, Manufacturer, City, State, Country).
- Response
We appreciate your kind comments. According to the mentioned suggestion, we separated the materials and methods in the text as below sections,
2.1 Preparation of the HA/SA bio-ink
2.2 Fabrication and characterization of the HA/SA scaffold
2.3 Cytocompatibility of the cultured cells in HA/SA scaffold
- Suggestion 4
It is needed to insert FTIR spectra of used hyaluronic acid and sodium alginate before 3D bio-printing in Figure 3, as well as FTIR spectra of bio-ink before and after crosslinking, for better comparison and analysis results of obtained blends.
- Response
First of all, we have corrected all the wrong expression about the experimental group S60H40, and changed to S70H30 or deleted in the text. For further understand, it is important to compare between crosslinking properties as your kind comments. Because we carried out FTIR for just measure the chemical composition depending of ratio of HA. Please understand it and we will check the mentioned points in the further study. Once again, we apologize for the unobvious descriptions in the text.
- Suggestion 5
Could the authors provide data about the amount of unreacted hyaluronic acid and sodium alginate, which is also important for in vivo application?
- Response
We appreciate your kind comments. In this study, we focused on the manufacturing process to fabricate the scaffolds using bioprinting with HA and SA blended bio-ink. We speculate that point to the amount of unreacted HA is dependent to the scale for the scaffold for targeting tissue. Please understand it and we will check the mentioned points in the further in vivo study.
- Suggestion 6
Did the authors analyze swelling properties in different physiological fluids (e.g. at pH 7.0 and pH 7.4 and 37 °C)? The swelling ratio measurement will be useful for better characterization and possible future application in soft tissue regeneration.
- Response
We appreciate your kind comments. In this study, we cultured the printed scaffolds in the medium (pH 7.2-7.4) at 37 °C maintained incubator. As we mentioned in the manuscript, we expect that the crosslinked alginate could maintain their structure during culture time. Because the uncrosslinked HA could swell easily with the increased their ration in the bio-ink. Please understand it and we will check the mentioned points in the further study.
- Suggestion 7
Also, I would like to advise authors for additional improvement of this paper quality by adding morphological analysis of obtained bio-ink, lyophilized in swollen state, e.g. by using scanning electron microscope method.
- Response
We appreciate your kind comments. The morphological analysis is important point to their hydrogel networks. In this study, we focused on the manufacturing process to fabricate the scaffolds using bioprinting with HA and SA blended bio-ink. As shown in the figure 6 and 7, HA could enhance the bioactivity in the cells bio-ink. We will check the morphological analysis in the further study.
- Suggestion 8
All figures and table should be inserted in the main text according Instruction and Journal`s template.
- Response
First of all, we apologize for not following the format of the journal, such as not placing photos between texts. According to the reviewer’s suggestion, corrections of main text including the figures were carried out. Thanks for your kind comment.
- Suggestion 9
It is common for scientific papers to be written in passive. Please, avoid 1st person plural and rewrite all sentences to 3rd person plural (Lines: 14, 42, 44, 62, 110, 151, 155, 159 and 164). After revision by the authors, this manuscript can be considered to be published preferably as Short Communication in the journal Polymers.
- Response
First of all, we apologize for our lack of English expression. According to the reviewer’s suggestion, additional English corrections were carried out, and the part of deficiencies were revised. Thanks for your kind comment.

Reviewer 2 Report
The English written shall be carefully revised for a few small errors, like "method fabricate" instead of "method to fabricate" in the introduction; Verify the correct use of the terms: Bioink (Hydrogel + cells) and construct (Bioink printed in some geometry); How many layers height are the constructs and how many layers can be stacked without collapsing? Can be an important information. Increase citations about this sentence "In particular, all bio-inks exhibit a shear thinning behavior, in which the shear viscosity decreases with the increasing shear rate" The graphic in Figure 2B does not represent a shear thinning behavior. Please, comment this.Author Response
- Suggestion 1
The English written shall be carefully revised for a few small errors, like "method fabricate" instead of "method to fabricate" in the introduction;
- Response
First of all, we apologize for our lack of English expression. According to the reviewer’s suggestion, additional English corrections were carried out, and the part of deficiencies were revised. Thanks for your kind comment.
- Suggestion 2
Verify the correct use of the terms: Bioink (Hydrogel + cells) and construct (Bioink printed in some geometry);
- Response
First of all, we apologize for our lack of English expression. According to the reviewer’s suggestion, additional English corrections were carried out, and the part of deficiencies were revised. Thanks for your kind comment.
- Suggestion 3
How many layers height are the constructs and how many layers can be stacked without collapsing?
- Response
We appreciate your kind comments. All scaffolds were stacked without collapsing during printing process. Because the alginate was pre-crosslinked with Ca2+, the structure could stack more 2 mm height with 4 layers. The structure was designed at 1.5 mm height after crosslinking process.
- Suggestion 4
Can be an important information. Increase citations about this sentence "In particular, all bio-inks exhibit a shear thinning behavior, in which the shear viscosity decreases with the increasing shear rate" The graphic in Figure 2B does not represent a shear thinning behavior. Please, comment this.
- Response
We appreciate your kind comments. As your comment, it was unclear to show the shear thinning behavior because they have tested in wide range of the shear rate. Therefore, we revised the data image for better understand to the reader at Figure 2B. Thanks for your kind comment.

Reviewer 3 Report
The manuscript “Three-Dimensional Printable Hydrogel using a Hyaluronic Acid/Sodium Alginate Bio-ink” deals with the production of biocompatible hydrogels of hyaluronic acid and sodium alginate, without chemical crosslinking, for tissue engineering applications. The work is interesting and good results have been obtained. However, major revisions have to be performed before the publication.
In particular:
- Abstract. Please, add some quantitative results.
- Introduction. The state of art is completely missing. Please, add some work about the use of sodium alginate and hyaluronic acid in tissue engineering, with the aim of highlighting the novelty of the present work. Some useful papers can be, for instance: Baldino et al., A new tool to produce alginate-based aerogels for medical applications, by supercritical gel drying, Journal of Supercritical Fluids, 2019, 146, pp. 152-158; Liu et al., Polymeric hybrid aerogels and their biomedical applications, Soft Matter, 2020, 16, pp. 9160-9175; etc…
- Materials and Methods. Please, separate all materials (with all their characteristics) from the preparation methods.
- English should be improved.
Author Response
- Suggestion 1
Abstract. Please, add some quantitative results.
- Response
We appreciate your kind comments. According to the mentioned suggestion, we have included more quantitative results and descriptions in the text as below,
“Bio-ink properties have been extensively studied for use in the three-dimensional (3D) bio-printing process for tissue engineering applications. In this study, we developed a method to synthesize bio-ink using hyaluronic acid (HA) and sodium alginate (SA) without employing the chemical crosslinking agents of HA to 30% (w/v). Furthermore, we evaluated the properties of the obtained bio-inks to gauge their suitability in bio-printing, primarily focusing on their viscosity, printability, and shrinkage properties. Furthermore, the bio-ink encapsulating the cells (NIH3T3 fibroblast cell line) was characterized using a live/dead assay and WST-1 to assess the biocompatibility. It was inferred from the results that the blended hydrogel was successfully printed for all groups with viscosities of 883 Pa∙s (HA, 0% w/v), 1211 Pa∙s (HA, 10% w/v), and 1525 Pa∙s, (HA, 30% w/v) at a 0.1 s−1 shear rate. Their structures exhibited no significant shrinkage after CaCl2 crosslinking and maintained their integrity during the culture periods. The relative proliferation rate of the encapsulated cells in the HA/SA blended bio-ink was 70% higher than the SA-only bio-ink after the fourth day. These results suggest that the 3D printable HA/SA hydrogel could be used as the bio-ink for tissue engineering applications.”
- Suggestion 2
Introduction. The state of art is completely missing. Please, add some work about the use of sodium alginate and hyaluronic acid in tissue engineering, with the aim of highlighting the novelty of the present work. Some useful papers can be, for instance: Baldino et al., A new tool to produce alginate-based aerogels for medical applications, by supercritical gel drying, Journal of Supercritical Fluids, 2019, 146, pp. 152-158; Liu et al., Polymeric hybrid aerogels and their biomedical applications, Soft Matter, 2020, 16, pp. 9160-9175; etc…
- Response
We appreciate your kind comments. According to the mentioned suggestion, the sentences have been added to read in the text as below,
“As a natural polysaccharide, sodium alginate (SA) has been widely used as the base material in bio-printing, owing to its biocompatibility, biodegradability, and viscoelasticity to facilitate cell transplantation by printing [6]. However, SA cannot interact with mammalian cells, owing to its lack of cell adhesion moieties, which is an important requirement for cellular functions [7]. Therefore, studies have proposed that designing the bio-ink by blending various materials can potentially promote cellular activity in bio-inks comprising SA [8, 9]. Recently, novel tools employing hyaluronic acid (HA) with SA to provide an improved environment to cells were extensively investigated for medical applications [10-12].”
References
[10] Baldino, L.; Cardea, S.; Scognamiglio, M.; Reverchon, E. A new tool to produce alginate-based aerogels for medical applications, by supercritical gel drying. J. Supercrit. Fluids 2019, 146, 152–158; doi:10.1016/j.supflu.2019.01.016.
[11] Liu, Z.; Ran, Y.; Xi, J.; Wang, J. Polymeric hybrid aerogels and their biomedical applications. Soft Matter 2020, 16, 9160–9175; doi:10.1039/d0sm01261
- Suggestion 3
Materials and Methods. Please, separate all materials (with all their characteristics) from the preparation methods.
- Response
We appreciate your kind comments. According to the mentioned suggestion, we separated the materials and methods in the text as below,
2.1 Preparation of the HA/SA bio-ink
SA was purchased from Sigma-Aldrich. HA of 2,500 kDa was supplied from Humedix.Co. (Korea). HA and alginate were dissolved in high glucose-DMEM (Dulbecco’s Modified Eagle’s Medium) with 1% penicillin/streptomycin and physically mixed in different ratio (w/v) in a final concentration of 0%, 10% and 30% (w/v), respectively. The group names and compositions of the hydrogels were shown in Table 1. The hydrogels were prepared with three different HA/SA volume ratios. For making the bio-ink, the NIH3T3 cells were cultured in high glucose-DMEM supplemented with 10% fetal bovine serum and 1 % penicillin/streptomycin. These cultures were maintained in the incubator at 37℃ and 5 % CO2. The cell encapsulated bio-ink gently mixed with HA/SA hydrogels to reach a final concentration of 1 x 106 cells/ml.
2.2 Fabrication and characterization of the HA/SA scaffold
As shown in figure 1, we used a 3D solid freeform fabrication (SFF) bio-printing system (laboratory made system in the Korea Institute of Machinery and Materials) [7]. We fabricated a cell-laden construct in a layer-by-layer fashion with the designed 20 x 20 x 1.5 mm3. HA/SA hydrogels were analyzed by the rheometer (TA instruments) at the room temperature to measure the viscosities of the hydrogels. The infrared spectra of HA/SA hydrogel was measured using an FT-IR spectrophotometer (Perkin-Elmer) measured spectrum condition at a resolution of 4.0 cm–1 over the range of 500–4,000 cm–1. For the frequency sweep experiments, frequency was varied from 1 to 10 Hz at 5% strain while for the strain sweep experiments. The printability of the HA/SA hydrogels were evaluated by bio-printing system. To assess shrinkage behavior, the hydrogel printed structures were measured for size variation before and after treatment with CaCl2. The shrinking rate analysis of structure area was carried out using an image J program (NIH, USA).
- Suggestion 4
English should be improved.
- Response
First of all, we apologize for our lack of English expression. According to the reviewer’s suggestion, additional English corrections were carried out, and the part of deficiencies were revised. Thanks for your kind comment.

Reviewer 4 Report
The goal of this study was to assess the physical and biological properties of a bio-ink based on sodium alginate and hyaluronic acid.
The research topic is interesting and can provide an insight about new printable hydrogel combinations to use in regenerative medicine.
However, there are some concerns that should be addressed to solve some limitations of the present manuscript:
The introduction must provide a better background of the research progression in this field. The flow of the writing needs to be improved and expand more about previous research to improve the properties of alginate composite bio-inks, including reference to
https://doi.org/10.1007/s10856-020-06440-3
and other recent papers on the topic. A better definition of current biomedical applications of hyaluronic acid, and potentially new fields, including regenerative endodontic procedures
http://dx.doi.org/10.1016/j.joen.2017.03.005
must be addressed.
The introduction must end with a null hypothesis to be tested in this research.
Legend of figure 3 needs to be improved, to be self-explanatory.
P3L112, there is no such a combination as S60H40 analyzed in Figure 3 or in the present research. All this paragraph needs to be re-written, because data from the present study is mixed up with data from previous studies in a vary unclear and confusing form.
Data from Figure 7 is hard to interpret in a comprehensible and transparent analysis. The experiment should include a positive control group, to assess the proliferation of cells in an appropriate culture medium, and data from the groups of the tested bio-inks should be presented as a relative percentage of the positive control group proliferation.
Author Response
- Suggestion 1
The introduction must provide a better background of the research progression in this field. The flow of the writing needs to be improved and expand more about previous research to improve the properties of alginate composite bio-inks, including reference to https://doi.org/10.1007/s10856-020-06440-3 and other recent papers on the topic. A better definition of current biomedical applications of hyaluronic acid, and potentially new fields, including regenerative endodontic procedures http://dx.doi.org/10.1016/j.joen.2017.03.005 must be addressed.
- Response
We appreciate your kind comments. According to the mentioned suggestions, the sentences have been added to read in the text as below,
“Therefore, studies have proposed that designing the bio-ink by blending various materials can potentially promote cellular activity in bio-inks comprising SA [8, 9]. Recently, novel tools employing hyaluronic acid (HA) with SA to provide an improved environment to cells were extensively investigated for medical applications [10-12]. HA, a glycosaminoglycan that acts as a signaling molecule for cell migration and proliferation, has been extensively used as a hydrogel material in biomedical applications [13, 14]. However, it is hard to design a controllable structure with the desired shape, porosity, and interconnected pores in the scaffold for the regeneration of large defects using conventional fabrication methods. Therefore, a method for fabricating the scaffold using blended hydrogel and 3D bio-printing needs to be investigated.”
References
[8] Wei, L.; Li, Z.; Li, J.; Zhang, Y.; Yao, B.; Liu, Y.; Song, W.; Fu, X.; Wu, X.; Huang, S. An approach for mechanical property optimization of cell-laden alginate–gelatin composite bioink with bioactive glass nanoparticles. J. Mater. Sci. Mater. Med. 2020, 31,103; doi:10.1007/s10856-020-06440-3.
[9] Ahmad Raus, R.; Wan Nawawi, W.M.F.; Nasaruddin, R.R. Alginate and alginate composites for biomedical applications. Asian J. Pharm. Sci. 2021; doi:10.1016/j.ajps.2020.10.001.
Baldino, L.; Cardea, S.; Scognamiglio, M.; Reverchon, E. A new tool to produce alginate-based aerogels for medical applications, by supercritical gel drying. J. Supercrit. Fluids 2019, 146, 152–158; doi:10.1016/j.supflu.2019.01.016.
[12] Palma, P.J.; Ramos, J.C.; Martins, J.B.; Diogenes, A.; Figueiredo, M.H.; Ferreira, P.; Viegas, C.; Santos, J.M. Histologic Evaluation of Regenerative Endodontic Procedures with the Use of Chitosan Scaffolds in Immature Dog Teeth with Apical Periodontitis. J. Endod. 2017, 43, 1279–1287; doi:10.1016/j.joen.2017.03.005.
- Suggestion 2
The introduction must end with a null hypothesis to be tested in this research.
- Response
We appreciate your kind comments. According to the mentioned suggestion, the sentences have been added to read in the text as below,
“In this study, we investigated a biologically enhanced bio-ink using HA, which could improve the cellular functionality of the bio-ink using the alginate alone for 3D bio-printing. We anticipate that SA can support the printability of the bio-ink, while HA enhances the bioactivity in the encapsulated cells.”
- Suggestion 3
Legend of figure 3 needs to be improved, to be self-explanatory. P3L112, there is no such a combination as S60H40 analyzed in Figure 3 or in the present research. All this paragraph needs to be re-written, because data from the present study is mixed up with data from previous studies in a vary unclear and confusing form.
- Response
First of all, we apologize for the confusing expressions shown in figure 3 and in the text. As your kind comment, the S60H40 group was excluded in the experimental group because it was difficult to conduct the experiment by their unstable structural maintain properties during the culture. Therefore, we have corrected all the wrong expression about the experimental group S60H40, and changed to S70H30 or deleted in the text. Once again, we apologize for the unobvious descriptions in the text.
- Suggestion 4
Data from Figure 7 is hard to interpret in a comprehensible and transparent analysis. The experiment should include a positive control group, to assess the proliferation of cells in an appropriate culture medium, and data from the groups of the tested bio-inks should be presented as a relative percentage of the positive control group proliferation.
- Response
First of all, we apologize for the uncomprehensive expressions shown in figure 7 and in the text. As your kind comment, we exchanged the experimental data to relative percentage with a control group using 100H0 at Day 1 as shown below,

Round 2
Reviewer 3 Report
The authors performed all the modifications proposed by the Reviewer.
Reviewer 4 Report
The authors performed the asked corrections and the manuscript improved.